# Distinct contributions of the thin and thick filaments to length-dependent activation in heart muscle

Xuemeng Zhang[1,2], Thomas Kampourakis[1,2], Ziqian Yan[1,2], Ivanka Sevrieva[1,2], Malcolm Irving[1,2], Yin-Biao Sun[1,2]*

[1]Randall Division of Cell and Molecular Biophysics, King's College London, London, United Kingdom; [2]British Heart Foundation Centre of Research Excellence, King's College London, London, United Kingdom

**Abstract** The Frank-Starling relation is a fundamental auto-regulatory property of the heart that ensures the volume of blood ejected in each heartbeat is matched to the extent of venous filling. At the cellular level, heart muscle cells generate higher force when stretched, but despite intense efforts the underlying molecular mechanism remains unknown. We applied a fluorescence-based method, which reports structural changes separately in the thick and thin filaments of rat cardiac muscle, to elucidate that mechanism. The distinct structural changes of troponin C in the thin filaments and myosin regulatory light chain in the thick filaments allowed us to identify two aspects of the Frank-Starling relation. Our results show that the enhanced force observed when heart muscle cells are maximally activated by calcium is due to a change in *thick* filament structure, but the increase in calcium sensitivity at lower calcium levels is due to a change in *thin* filament structure.

*For correspondence: yin-biao. sun@kcl.ac.uk

**Competing interests:** The authors declare that no competing interests exist.

## Introduction

The Frank-Starling law of the heart describes the relationship between cardiac stroke volume and end-diastolic volume and operates on a beat-to-beat basis. At the cellular level, the Frank-Starling relationship implies that increasing cardiac sarcomere length results in enhanced performance of cardiac muscle cells during the subsequent contraction (*Allen and Kentish, 1985*; *de Tombe et al., 2010*). This myofilament length-dependent activation (LDA) is two-fold: with an increase in sarcomere length (SL), there are increases in (1) the maximum force developed by the myofilaments at high $Ca^{2+}$ and (2) their sensitivity to $Ca^{2+}$. Impaired LDA and hence a defective Frank–Starling relationship may also contribute to the pathogenesis of hypertrophic cardiomyopathy and heart failure (*Schwinger et al., 1994*; *Sequeira et al., 2013*).

Although the Frank-Starling law of the heart is one of its fundamental properties and has been investigated for over a century, the detailed molecular mechanism for the LDA in heart muscle remains elusive. Over the last forty years, the main focus has been on the myofilament $Ca^{2+}$-sensitivity component of LDA.

Contraction of cardiac muscle is driven by an interaction between myosin and actin that is controlled by the transient binding of $Ca^{2+}$ ions to troponin in the actin-containing thin filament on a beat-to-beat basis (*Tobacman, 1996*; *Gordon et al., 2000*; *Kobayashi and Solaro, 2005*). An increase in SL leads to a decreased inter-filament spacing (*Irving et al., 2000*; *Lee et al., 2013*) and it has been suggested that this leads directly to an increase in $Ca^{2+}$ sensitivity of force generation (*McDonald and Moss, 1995*; *Fuchs and Smith, 2001*). However, when osmotic compression of inter-filament spacing matches that induced by SL, the myofilament $Ca^{2+}$ sensitivity is only affected

**eLife digest** The heart needs to pump out the same volume of blood that enters it. This is not as simple as it sounds, as changes in heart rate – for example, in response to exercise – alter how hard the heart must pump.

When blood flows into the heart it stretches the heart muscle, which consists of units called sarcomeres. Sarcomeres contain two types of protein filament, known as thick filaments and thin filaments. When a heartbeat is triggered by calcium ions flowing into the heart muscle cells, the thick filaments slide over the thin filaments. This causes the heart muscle cell to contract.

The Frank–Starling mechanism helps to regulate the contraction of the heart. This mechanism has two aspects. Firstly, as the sarcomere lengthens, its protein filaments are able to contract with more force for a given high level of calcium ions. Secondly, the lengthening of the sarcomere makes the filaments more sensitive to calcium ions, which again causes the heart to contract more forcefully. However, the molecular mechanisms that underlie these effects were not clear.

Zhang et al. have now studied rat heart muscle cells using a new fluorescence-based method that can detect structural changes in the thick and thin filaments. The results show that the increased force that is generated when sarcomeres are stretched can be accounted for by changes in the structure of the thick filament. In contrast, the increase in calcium sensitivity that occurs as the sarcomere lengthens is largely due to structural alterations in the thin filament. These two processes can be controlled independently, but work together in the Frank–Starling mechanism.

Now that we better understand the molecular basis of the Frank–Starling mechanism, further work could investigate new strategies for designing and testing treatments for heart disease.

by SL (*Konhilas et al., 2002*), indicating that the reduction of inter-filament spacing is unlikely to mediate LDA.

Titin-based passive tension in the heart muscle cells seems to play an important role in LDA (*Cazorla et al., 2001*; *Fukuda et al., 2001*; *Lee et al., 2010*). Such a mechanism might involve a strain sensor and a signal transduction pathway that conveys the strain signal to the contractile machinery. A single titin molecule runs from the Z-disc to the M-line in the A-band contributing to muscle assembly and passive tension (*Labeit and Kolmerer, 1995*). The fact that titin is largely responsible for the length-dependent passive tension of cardiac muscle and also possesses both actin and myosin binding domains (*Linke et al., 2002*; *Granzier and Labeit, 2004*), makes it a candidate for the link between SL and force generating apparatus. Small-angle X-ray diffraction studies suggest that length-dependent changes in the myosin head orientation in diastole may be a factor in LDA (*Farman et al., 2011*). Moreover, a recent X-ray study shows that length-dependent structural changes are also observed in the thin filament in diastole (*Ait-Mou et al., 2016*), supporting the notion that thin filament activation may also play an important role in LDA. The length-dependent changes in the structure of both thick and thin filaments are related to the titin strain (*Ait-Mou et al., 2016*).

To identify the molecular mechanism(s) underlying LDA, we measured the structural changes in the thin and thick filaments of intact sarcomeres in heart muscle induced by SL changes using bifunctional rhodamine (BR) probes on the cardiac troponin C (cTnC) and myosin regulatory light chain (cRLC) (*Sun et al., 2009*; *Kampourakis et al., 2014*). These probes allowed us to differentiate the structural changes in both types of filament caused by calcium activation, force-generating myosin heads and SL changes in the native environment of the cardiac sarcomere. The results presented here show that LDA is accompanied by distinctive structural changes in both the thin and thick filaments.

# Results

## Increase of maximum active force at longer SL is not associated with increased thin filament activation

$Ca^{2+}$-activated force development was determined in demembranated ventricular trabeculae from rat heart. Maximum active force at pCa 4.5 increased by ~40% when SL was increased from 1.9 to 2.3 μm (*Figure 1*). Passive force increased in the same SL range by ~30% of maximum $Ca^{2+}$-activated force at SL 1.9 μm (*Figure 1a*). Increasing SL also produced a leftward shift of the $Ca^{2+}$-force relationships, indicating an increase in myofilament $Ca^{2+}$ sensitivity (*Figure 1b*). These length-dependent changes in force are consistent with previous reports (*Kentish et al., 1986*; *Dobesh et al., 2002*).

Contraction of heart muscle is initiated by $Ca^{2+}$ binding to the regulatory domain of troponin in the actin-containing thin filaments, that triggers a cascade of structural changes in the thin filament and leads to an azimuthal movement of tropomyosin around the filament that allow myosin to interact with actin and generate force (*Tobacman, 1996*; *Gordon et al., 2000*; *Kobayashi and Solaro, 2005*). The IT arm of troponin, a rigid domain containing the C-terminal lobe of TnC and a coiled-coil formed by α-helices from troponin I (TnI) and troponin T (TnT), anchors the troponin complex to the thin filament. To determine whether the changes in SL-dependence of force are accompanied by structural changes in the thin filaments of cardiac muscle cells, we measured the orientation of the regulatory head and IT arm of troponin. Cardiac troponin C with bifunctional rhodamine (BR) cross-linking residues 55 and 62 along the C helix (BR-cTnC-C) in the regulatory head or 95 and 102 along the E helix (BR-cTnC-E) in the IT arm of troponin was incorporated into demembranated trabeculae (*Sun et al., 2009*). The polarized fluorescence intensities from trabeculae containing BR-labelled cTnC were used to calculate the order parameter $<P_2>$ that describes the orientation of the cTnC

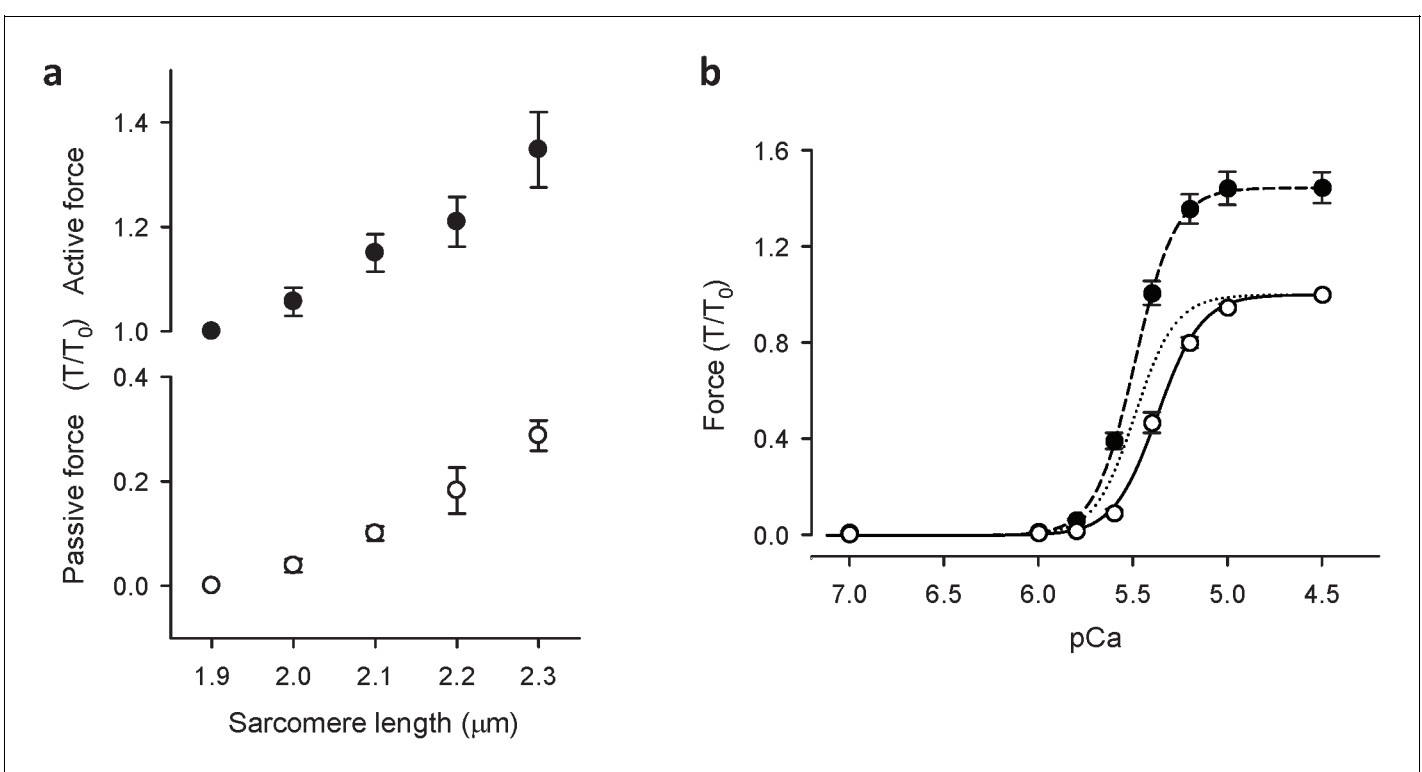

**Figure 1.** Sarcomere length-force relationships in rat cardiac trabeculae. (**a**) Passive force (○) and maximum $Ca^{2+}$-activated force (●) at five sarcomere lengths (n = 6 trabeculae). Force is normalised to maximum force measured at SL 1.9 μm ($T/T_0$). (**b**) pCa-force relationships at sarcomere lengths 1.9 μm (○) and 2.3 μm (●). The data were summarised from experiments for *Figure 2a and b* and fit to the Hill Equation (n = 10). Dotted line is the Hill curve at SL 2.3 μm normalised to its maximum force. Increases in SL resulted in increases in maximum $Ca^{2+}$-activated force and Ca sensitivity (leftward shift of pCa-force curve). Error bars denote SEM.

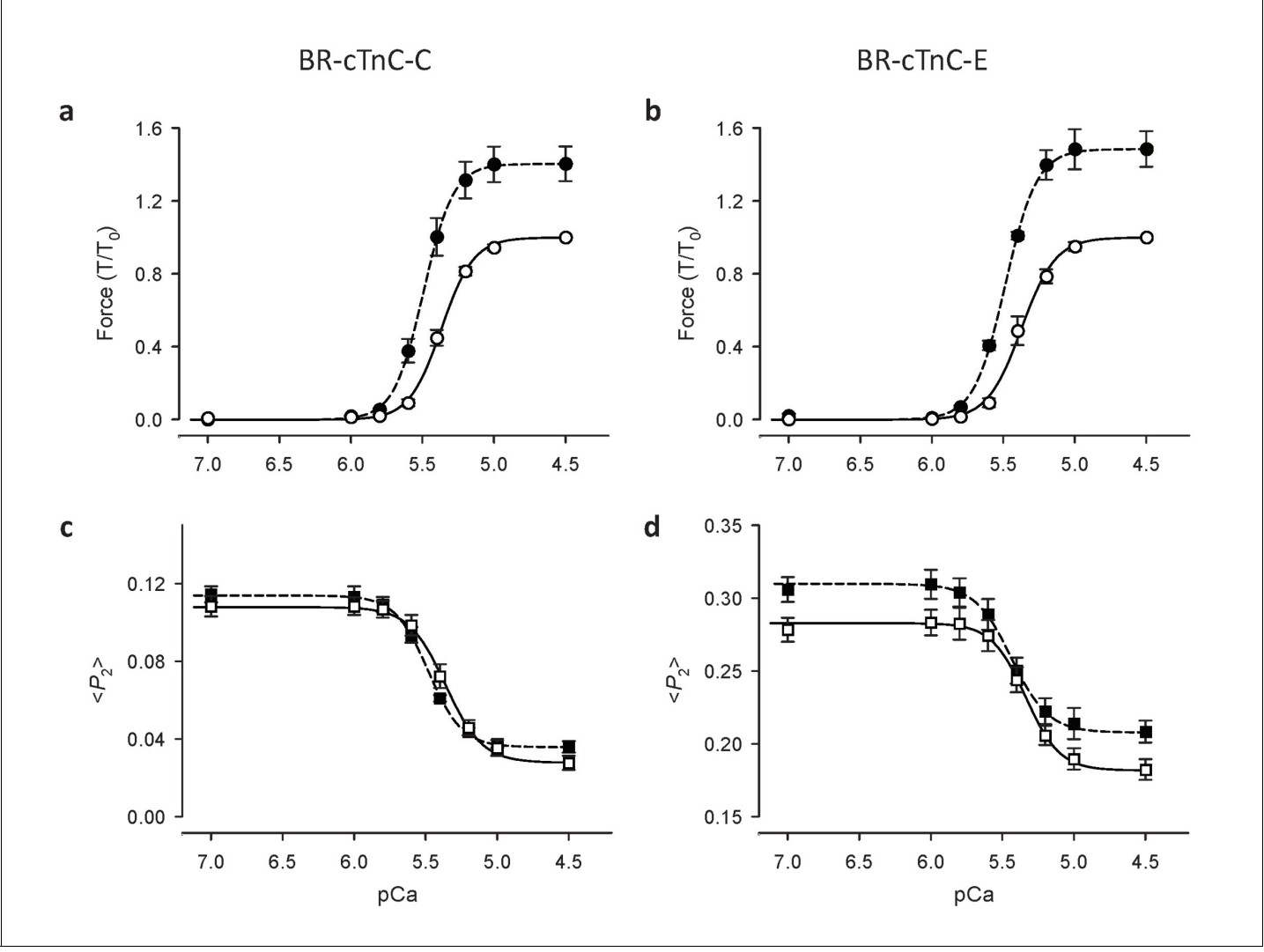

**Figure 2.** $Ca^{2+}$-dependence of force and troponin orientation, $<P_2>$, in cardiac trabeculae. (**a, c**) BR-cTnC-C; (**b, d**) BR-cTnC-E. Open symbols, SL 1.9 μm; filled symbols, SL 2.3 μm. Error bars denote SEM.

helix to which BR probe is attached with respect to the trabecular axis (for details, see Materials and methods).

The $Ca^{2+}$-dependence of the $<P_2>$ changes was described using the Hill equation, which gave a good fit to the data for both the C and E helix probes in the range of $[Ca^{2+}]$ between pCa 7 and 4.5 (*Figure 2*). Here $pCa_{50}$, the pCa for half-maximal changes in $<P_2>$, is a measure of $Ca^{2+}$ sensitivity; $n_H$, the Hill coefficient, describes the steepness of the $Ca^{2+}$ dependence and is a measure of the cooperativity of the $Ca^{2+}$-dependent change.

## Length-dependent orientational changes of the C helix of cTnC

The orientation of the cTnC C helix in the regulatory domain of troponin was determined using a cTnC mutant with BR cross-linking residues 55 and 62 (BR-cTnC-C). At a short SL of 1.9 μm, with the increase of active force, $<P_2>$ for BR-cTnC-C decreased by an average of $-0.081 \pm 0.005$ between pCa 6 and 4.5, the range of $[Ca^{2+}]$ where active force was developed (*Figure 2c*, *Table 1*). The normalized changes in $<P_2>$ and force had a very similar $Ca^{2+}$ dependence. The $pCa_{50}$ for $<P_2>$ was $5.37 \pm 0.02$, almost identical to that for force in the same trabeculae (*Table 1*).

**Table 1.** Ca$^{2+}$-dependence of force and the orientation parameter $<P_2>$ and the effects of sarcomere length. Mean ± SEM. pCa$_{50}$ and $n_H$ are fitted parameters of Hill equation. $\Delta<P_2>$ , changes in $<P_2>$ during Ca$^{2+}$-activation from pCa 6 to 4.5. Comparisons: between sarcomere lengths 1.9 and 2.3 μm ($t$ test, two-tailed; *p<0.05).

| | BR-cTnC-C | | BR-cTnC-E | |
|---|---|---|---|---|
| SL (μm) | 1.9 | 2.3 | 1.9 | 2.3 |
| Force | | | | |
| (mN/mm$^2$) | 22.7 ± 1.8 | 32.4 ± 1.5* | 23.2 ± 3.5 | 33.1 ± 6.0* |
| pCa$_{50}$ | 5.37 ± 0.03 | 5.50 ± 0.03* | 5.37 ± 0.02 | 5.49 ± 0.03* |
| $n_H$ | 4.03 ± 0.20 | 4.29 ± 0.22* | 4.12 ± 0.24 | 4.33 ± 0.16* |
| $<P_2>$ | | | | |
| pCa$_{50}$ | 5.37 ± 0.02 | 5.48 ± 0.01* | 5.34 ± 0.02 | 5.45 ± 0.02* |
| $n_H$ | 3.28 ± 0.16 | 3.42 ± 0.10 | 3.59 ± 0.25 | 3.24 ± 0.18 |
| at pCa 6 | 0.108 ± 0.005 | 0.113 ± 0.005 | 0.283 ± 0.009 | 0.310 ± 0.010* |
| at pCa 4.5 | 0.028 ± 0.004 | 0.036 ± 0.003* | 0.182 ± 0.007 | 0.208 ± 0.008* |
| $\Delta<P_2>$ | −0.081 ± 0.005 | −0.077 ± 0.005 | −0.101 ± 0.007 | −0.101 ± 0.006 |
| | n = 5 | | n = 5 | |

When trabeculae were stretched from SL 1.9 to 2.3 μm, the maximal Ca$^{2+}$-activated force from trabeculae containing BR-cTnC-C increased by about 40% (*Figure 2a* and *3a*, *Table 1*), and the pCa$_{50}$ increased by 0.12 ± 0.01 (n = 5) pCa units, indicating an increase in Ca$^{2+}$-sensitivity of force generation. The pCa$_{50}$ for $<P_2>$ of the BR-cTnC-C probe also increased by 0.11 ± 0.02 pCa units, similar to that for force. In contrast with the effect on maximum Ca$^{2+}$-activated force, the amplitude of the change in $<P_2>$ ($\Delta<P_2>$ in *Table 1*) for BR-cTnC-C between pCa 6 and 4.5 was unaffected by SL increase (*Figure 2c and 3c*, *Table 1*). The steepness of the Ca$^{2+}$-dependent changes for $<P_2>$, $n_H$, also did not change significantly when the SL was increased (*Table 1*).

## Length-dependent orientational changes of the E helix of cTnC

The orientation of the cTnC E helix in the IT arm of troponin was determined using a cTnC mutant with BR cross-linking residues 95 and 102 (BR-cTnC-E). Upon Ca$^{2+}$ activation at SL 1.9 μm, the order parameter $<P_2>$ decreased during calcium activation between pCa 6 and 4.5. pCa$_{50}$ for $<P_2>$ of the cTnC E helix probe at 1.9 μm SL was 5.34 ± 0.02, not significantly different from pCa$_{50}$ for force in the same group of trabeculae (*Table 1*). Increasing SL from 1.9 to 2.3 μm significantly increased pCa$_{50}$ for $<P_2>$ and force by 0.11 and 0.12 pCa units, respectively, the same as for the C helix probe. Similar to that for BR-cTnC-C, the changes in $<P_2>$ for the C helix probe during Ca$^{2+}$-activation ($\Delta<P_2>$) were not affected by increasing SL from 1.9 to 2.3 μm (*Table 1*).

For both the C and E helix probes, $<P_2>$ decreased upon Ca$^{2+}$-activation and the overall changes in $<P_2>$ ($\Delta<P_2>$) were not affected by increasing SL from 1.9 to 2.3 μm while the maximal force generation was markedly enhanced (*Figure 2 and 3* and *Table 1*). By contrast, increasing SL increased $<P_2>$ during both relaxation and active contraction, particularly for the BR-cTnC-E, i.e. the orientation change was in the opposite direction to that during Ca$^{2+}$-activation (*Figure 2d* and *Table 1*). These two observations together suggested that the activation of the thin filament at maximum [Ca$^{2+}$] (pCa 4.5) was not changed by increasing SL. The increased $<P_2>$ at longer SL at all [Ca$^{2+}$] may be related to the fact that ventricular trabeculae are made up of short branched cardiac muscle cells in which the myofibrils are not perfectly parallel. The myofibrils are likely to become better aligned when trabeculae are stretched, and this would contribute to the higher $<P_2>$ at longer SL.

These results show that the effect of an increase in SL on the Ca$^{2+}$ sensitivity of structural changes in the thin filaments reported by the cTnC probes is the same as that on force, but the higher maximum active force at longer SL is not accompanied by greater activation of the thin filament at maximum calcium activation. These results suggest that the thin filament may already be fully activated

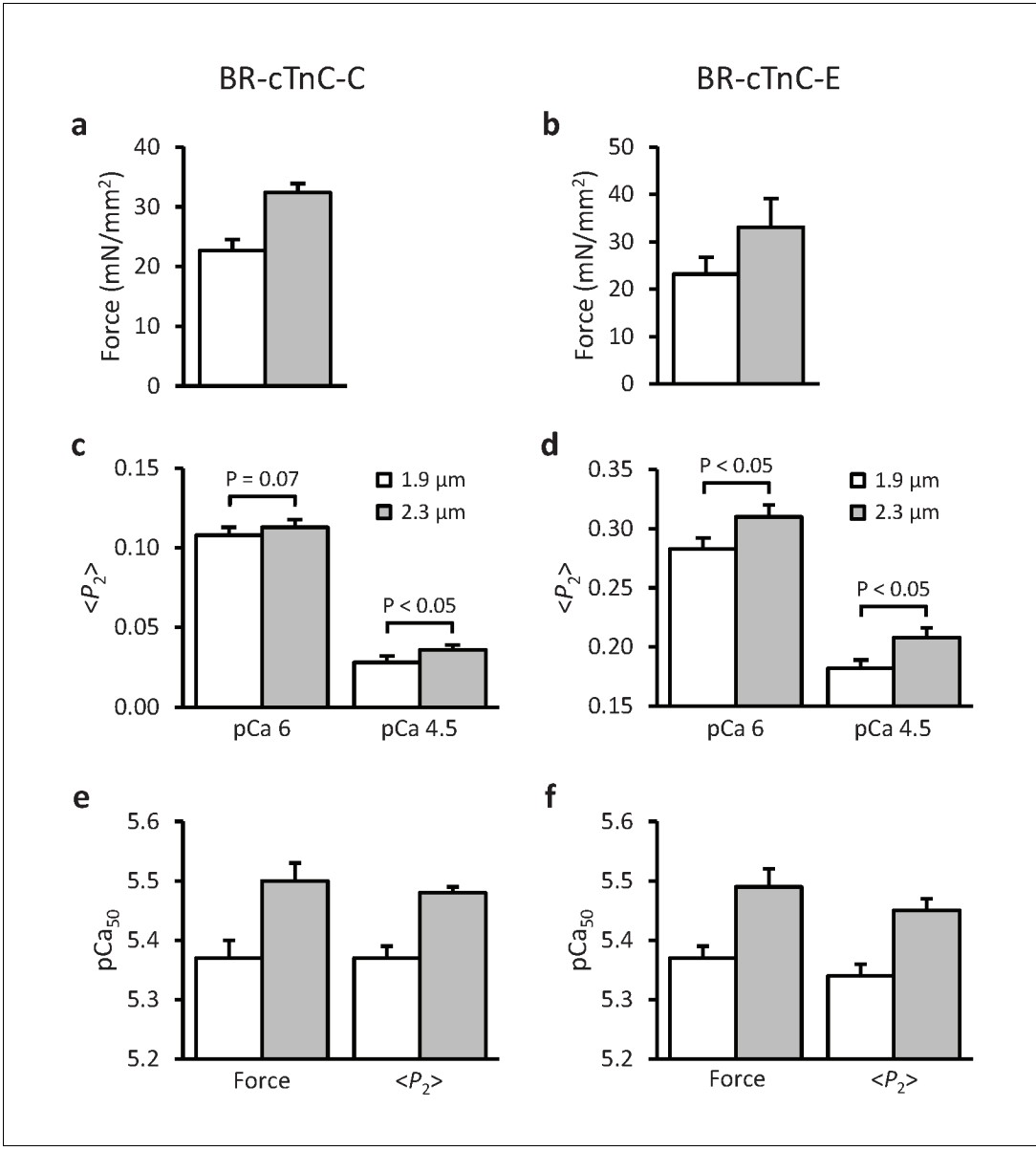

**Figure 3.** Effects of sarcomere length on maximum $Ca^{2+}$-activated force, orientation ($<P_2>$) of cTnC-BR probes in solutions with the highest $[Ca^{2+}]$ where trabeculae remained relaxed (pCa 6) and the maximum $[Ca^{2+}]$ (pCa 4.5), and the Hill parameter $pCa_{50}$. (**a, c, e**) BR-cTnC-C; (**b, d, f**) BR-cTnC-E. Open bars, SL 1.9 µm; grey bars, SL 2.3 µm. Mean ± SEM (n = 5). The statistical significance of differences was assessed using a two-tailed paired *t*-test.

at pCa 4.5 in control conditions at short SL, and that the length-dependent increase in maximum active force is determined by processes downstream of thin filament activation in the signalling pathway.

## Length-dependent Ca²⁺ sensitivity of cTnC structural changes is independent of the presence of force-generating myosin heads

It is generally thought that force-generating myosin heads sensitise the thin filament to calcium (*Moss et al., 2004*; *Smith et al., 2009*), and this led to the idea that LDA might be a consequence of the higher force or number of myosin heads attached to actin at longer SL. We tested the role of force-generating myosin heads in the length-dependent structural changes of cTnC by inhibiting force generation using blebbistatin (*Straight et al., 2003*). In the presence of 25 µM blebbistatin,

the active force was reduced to 1.5% ± 0.4% (n = 12) of the control value. It should be noted that blebbistatin may inhibit active force generation by stabilising the OFF structure of the myosin thick filament in which the myosin heads are folded back on the surface of the thick filaments and unavailable for interacting with actin in the thin filaments (*Zhao et al., 2008*; *Xu et al., 2009*). The $Ca^{2+}$-sensitivity and steepness of the cTnC C helix orientational changes were reduced after addition of blebbistatin at SL 1.9 μm (*Figure 4a*, short-dashed line; *Table 2*), and $pCa_{50}$ decreased by ~0.08 pCa units (*Figure 4c*). The change in $<P_2>$ associated with activation (from pCa 6.0 to 4.5; $\Delta<P_2>$) was ~16% smaller after addition of blebbistatin (p<0.05; *Table 2*), although the effect of blebbistatin at each of these pCa values was not significant at the 5% level. Similar results were obtained for the E-helix probe (*Figure 4b*), except that $<P_2>$ increased significantly on addition of blebbistatin at both pCa 6 and pCa 4.5 (*Table 2*). The addition of blebbistatin also reduced the steepness of the $Ca^{2+}$ dependence for both the C and E helix orientations ($n_H$, *Table 2*). Similar results were obtained previously (*Sun et al., 2009*; *Robertson et al., 2015*), indicating that either active force increases myofilament $Ca^{2+}$ sensitivity or stabilising the OFF structure of the thick filament decreases the $Ca^{2+}$ sensitivity.

Increasing SL from 1.9 to 2.3 μm after addition of blebbistatin increased $pCa_{50}$ by ~0.08 pCa units for both the C and E helix probes (*Figure 4c and d*, *Table 2*), similar to the ca 0.11 pCa unit increase

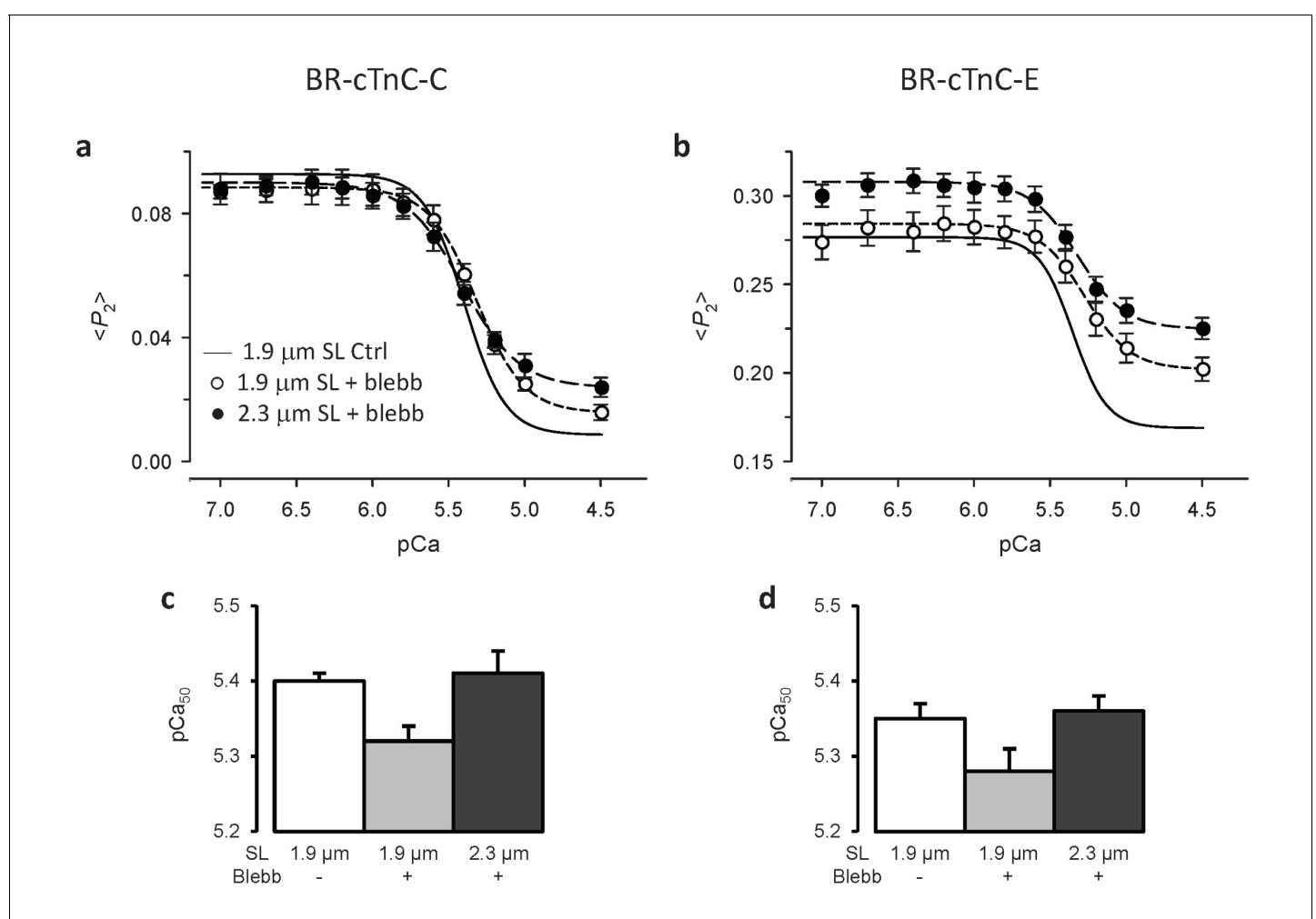

**Figure 4.** Effects of active force inhibition by 25 μM blebbistatin on the orientation of cTnC probes and their length-dependence. (**a, c**) BR-cTnC-C; (**b, d**) BR-cTnC-E. (**a–b**) Continuous lines denote Hill fits to data at SL 1.9 μm (data not shown for clarity). Circles denote $<P_2>$ in the presence of blebbistatin at SL of 1.9 μm (○) and 2.3 μm (●). (**c–d**) Fitted Hill parameter, $pCa_{50}$, for the control at 1.9 μm SL (white) and in the presence of blebbistatin at 1.9 μm SL (gray) and 2.3 μm (black). Mean ± SEM (n = 5–7).

**Table 2.** Effects of force inhibition by 25 µM blebbistatin on $Ca^{2+}$-dependence of force and the cTnC orientation parameter $<P_2>$. Mean ± SEM. $pCa_{50}$ and $n_H$ are fitted parameters of Hill equation. $\Delta<P_2>$, changes in $<P_2>$ during $Ca^{2+}$-activation from pCa 6 to 4.5. Comparisons (paired $t$ test, two-tailed): before and after addition of blebbistatin (*p<0.05); between sarcomere lengths 1.9 and 2.3 µm in the presence of blebbistatin ($^{\#}$p<0.05).

| | BR-cTnC-C | | | BR-cTnC-E | | |
|---|---|---|---|---|---|---|
| SL (µm) | 1.9 | 1.9 | 2.3 | 1.9 | 1.9 | 2.3 |
| 25 µM Blebbistatin | − | + | + | − | + | + |
| **Force** | | | | | | |
| $pCa_{50}$ | 5.39 ± 0.03 | | | 5.39 ± 0.04 | | |
| $n_H$ | 3.72 ± 0.23 | | | 4.23 ± 0.20 | | |
| $<P_2>$ | | | | | | |
| $pCa_{50}$ | 5.40 ± 0.01 | 5.32 ± 0.02 * | 5.41 ± 0.03 $^{\#}$ | 5.35 ± 0.02 | 5.28 ± 0.03 * | 5.36 ± 0.02 $^{\#}$ |
| $n_H$ | 3.29 ± 0.10 | 2.74 ± 0.20 * | 2.49 ± 0.15 | 3.57 ± 0.22 | 2.90 ± 0.19 * | 2.60 ± 0.17 |
| at pCa 6.0 | 0.091 ± 0.004 | 0.088 ± 0.005 | 0.086 ± 0.004 | 0.276 ± 0.010 | 0.282 ± 0.010 * | 0.305 ± 0.009 $^{\#}$ |
| at pCa 4.5 | 0.009 ± 0.002 | 0.016 ± 0.002 | 0.024 ± 0.003 | 0.168 ± 0.008 | 0.202 ± 0.007 * | 0.225 ± 0.006 $^{\#}$ |
| $\Delta<P_2>$ | 0.082 ± 0.004 | 0.069 ± 0.004 * | 0.062 ± 0.003 $^{\#}$ | 0.098 ± 0.007 | 0.080 ± 0.004 * | 0.080 ± 0.005 |
| | n = 5 | | | n = 7 | | |

observed before the addition of blebbistatin (p>0.05, two-tailed $t$-test). The decrease in $<P_2>$ for the C-helix probe associated with activation ($\Delta<P_2>$) was smaller at the longer SL in the presence of blebbistatin (p<0.05, **Table 2**). For the E helix probe, on the other hand, increasing SL changed $<P_2>$ over the whole range of $[Ca^{2+}]$ in a direction characteristic of lower activation, with no change in $\Delta<P_2>$ (**Figure 4b**). These results show that the length-dependent increase in the $Ca^{2+}$-sensitivity of cTnC structural changes is retained after complete force inhibition and stabilisation of the OFF structure of the thick filament. Thus, the SL seems to directly modulate thin filament activation, and the higher force/number of force-generating myosin heads at longer SL is NOT the main cause of higher $Ca^{2+}$ sensitivity in LDA.

## Increase in maximal force in LDA is linked to a change in thick filament structure

The results described above show that the higher active force at longer SL is not due to increased thin filament activation. We therefore tested the possibility that it might be due to a different structural or regulatory state of the thick filament using a probe on the RLC region of the myosin motors. The orientation of this probe has previously been shown to be sensitive to both the change in orientation of the myosin heads associated with force generation per se and to interventions like RLC phosphorylation that alter the regulatory or structural state of the thick filament (**Kampourakis et al., 2014, 2015, 2016**). This bifunctional sulforhodamine (BSR) probe cross-links helices B and C in the N-terminal lobe of the cardiac myosin regulatory light chain (BSR-cRLC-BC), and is roughly perpendicular to the long axis of the myosin head. $<P_2>$ for the BSR-cRLC-BC probe increased when the trabeculae were activated at SL 1.9 µm and was further increased by stretching to SL 2.3 µm (**Figure 5**). Although $<P_2>$ increases with SL at both low (pCa 6.6) and high $Ca^{2+}$ (pCa 4.5), the effect was larger at high $Ca^{2+}$ (**Figure 5**). These results suggest that the higher active force at longer SL is associated with length-dependent changes in the orientation of the myosin heads.

## Discussion

Despite many years of investigation, the molecular mechanism underlying the Frank-Starling law of the heart and its cellular correlate, length-dependent activation, is still unclear. The results presented above show that changes in muscle length produce distinct structural changes in the thin and thick filaments of heart muscle. Although the present results were obtained from steady state

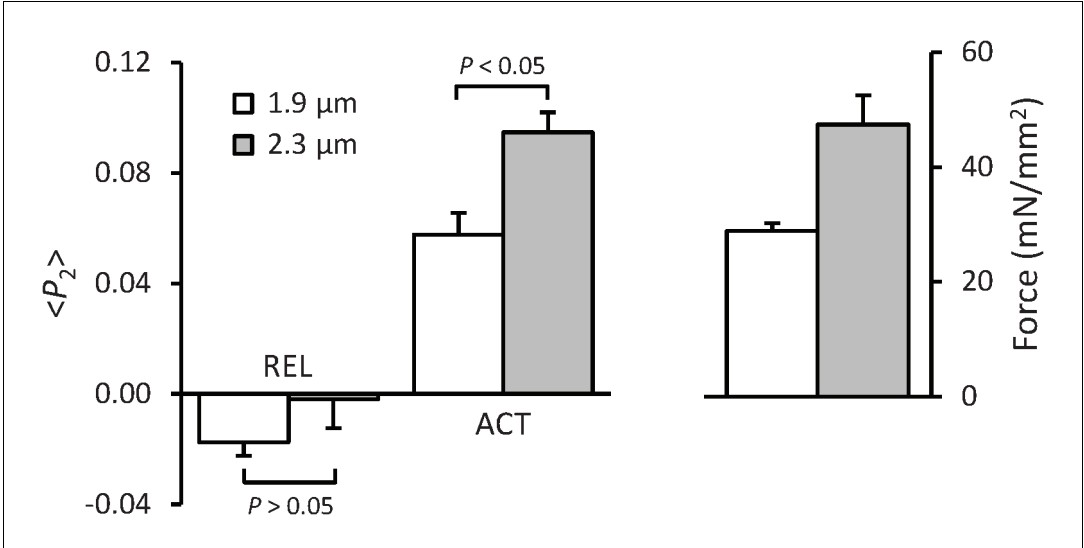

**Figure 5.** Force and orientation ($<P_2>$) of the BSR-RLC-BC probe in relaxing (pCa 6.6) and activating (pCa 4.5) solution at sarcomere lengths of 1.9 (white) and 2.3 μm (gray). Mean ± SEM (n = 5). Statistical significance was assessed using a two-tailed paired $t$-test.

measurements, they establish a molecular hypothesis that can be tested in future dynamic studies of LDA on the timescale of the heartbeat.

It has been shown previously that, due to the flexibility of the central linker between the D- and E-helices in cTnC, the conformation of the N-terminal domain of cTnC (NcTnC), the regulatory head of troponin, moves dynamically during activation (*Sevrieva et al., 2014*). Binding of $Ca^{2+}$ to NcTnC and its subsequent interaction with the switch region of cTnI initiate the removal of cTnI from its actin binding site by bending NcTnC towards the C-terminal domain of cTnC (*Takeda et al., 2003*; *Sevrieva et al., 2014*). Considering that the opening of NcTnC due to $Ca^{2+}$ binding and interaction with cTnI switch region is relatively small (*Dong et al., 1999*; *Li et al., 1999*), the C helix probe is likely to be much more sensitive to NcTnC re-orientation than to opening. Thus, the present results do not exclude the possibility that SL affects the opening of the NcTnC during cardiac muscle activation (*Li et al., 2014*). The E-helix probe (BR-cTnC-E) in the IT arm of troponin, on the other hand, is less likely to involve such multi-domain conformational changes since the IT arm acts as a scaffold holding NcTnC and the actin binding regions of cTnI in a fixed orientation on the thin filament (*Sevrieva et al., 2014*).

### Length-dependent structural changes in troponin are distinct from those induced by $Ca^{2+}$ activation or force generating myosin heads

Increasing SL induced a change in the orientation of both cTnC probes in the direction associated with lower activation (*Figure 2c and d*). In particular, the SL-dependent change in the orientation of the E helix probe in the IT arm of troponin occurred over the whole range of [$Ca^{2+}$], independent of the presence of force generating myosin heads (*Figure 4b*). These results show that the changes in troponin structure induced by increased SL are distinct from those induced by $Ca^{2+}$ activation or force generation.

A recent study by time-resolved small-angle X-ray diffraction also showed that increasing SL induces changes in troponin conformation in relaxed cardiac muscle cells (*Ait-Mou et al., 2016*). In addition, FRET measurements showed that the conformation of the switch region of cTnI relative to cTnC is also sensitive to SL in relaxed cardiac muscle (*Li et al., 2016*). These SL-dependent changes in troponin structure could be related to the increased passive force at long SL (*Figure 1a*). In cardiac muscle cells, the passive force is attributed to the giant protein titin, which runs from the Z-disc to the M-line (*Labeit and Kolmerer, 1995*; *Granzier and Labeit, 2004*; *Patel et al., 2012*). Titin-based passive force plays an important role in LDA (*Cazorla et al., 2001*; *Fukuda et al., 2001*,

*2003*). The length-dependent passive force is reduced in cardiac muscle from transgenic rats that express a giant splice isoform of titin, and the length-dependent change in the X-ray troponin reflection is also reduced in these muscle cells (*Ait-Mou et al., 2016*).

The combination of the present results with those of the X-ray and FRET studies mentioned above suggests that length-dependent changes in the structure of the thin filament of cardiac muscle can be one of the mechanisms underlying LDA, and hence the Frank-Starling law of the heart.

As discussed above, the increase in maximum active force in LDA is likely to be determined by processes downstream of thin filament activation in the signalling pathway. Our previous study described the SL-induced structural changes in the myosin filament over the whole range of $[Ca^{2+}]$ (*Kampourakis et al., 2016*). However, it made no direct comparison of $<P_2>$ for the BSR-cRLC-BC probe at different SLs as the primary focus of that work was on the effects of cRLC phosphorylation. The present results have shown that $<P_2>$ for the BSR-cRLC-BC probe increased during $Ca^{2+}$ activation and increased even further when SL was increased (*Figure 5*). Moreover, it has been shown that the length-dependent increase of isometric force in cardiac muscle is due to the change in the number of force-generating myosin heads (*Caremani et al., 2016*). Together, these results show that the higher active force at longer SL is associated with a more ON state of the thick filament.

X-ray measurements on resting cardiac muscle showed a close correlation between the orientation of myosin heads before activation and enhanced force generation as SL increases, further supporting the idea of a direct effect of SL on thick filament structure (*Farman et al., 2011*). Again, titin is implicated in the length-dependent structural changes in the thick filament. In resting cardiac muscles from rats that express the mutant titin described above, the SL-dependent changes in the intensities of the myosin-based X-ray reflections were greatly reduced, suggesting that titin-based passive force mediates the effect on thick filament structure (*Ait-Mou et al., 2016*).

## Dual filament regulation of length-dependent activation in cardiac muscle

It has long been established that an increase in SL in cardiac muscle results in increases in both the myofilament $Ca^{2+}$ sensitivity and the maximum $Ca^{2+}$-activated force (*Kentish et al., 1986*; *Dobesh et al., 2002*). The main focus has been on myofilament $Ca^{2+}$ sensitivity as a predominant component in LDA (*Allen and Kentish, 1985*) and myosin crossbridges as a main mediator (*Fitzsimons and Moss, 1998*; *Smith et al., 2009*). The present results provided evidence that two distinct pathways are involved in LDA. Increasing SL exerts a direct effect on the structure of troponin in the thin filament, which is the main contributor to the length-dependent increase of myofilament $Ca^{2+}$ sensitivity in LDA. A structural change of myosin in thick filament is responsible for the increase of maximal force generation in LDA.

It is well known that cTnI plays an important role in the LDA of cardiac muscle (*Arteaga et al., 2000*; *Konhilas et al., 2003*). It has also been shown that phosphorylation of Ser23/Ser24 at its cardiac-specific N-terminal region enhances the length-dependent increase in $Ca^{2+}$ sensitivity while having no effect on the length-dependent increase in maximum force (*Wijnker et al., 2014*). This is in support of our conclusion that the increase in $Ca^{2+}$ sensitivity in LDA is mediated through a change in thin filament structure. The thin filament mediated length-dependent change in $Ca^{2+}$ sensitivity may play an important role in the development of hypertrophic cardiomyopathy by missense mutations in the sarcomeric proteins (*Sequeira et al., 2013*).

The LDA occurs immediately after a length change and independently of the increase in the intracellular $[Ca^{2+}]$ (*Allen and Kurihara, 1982*; *Mateja and de Tombe, 2012*). Recent evidence has suggested that the passive force originated by titin plays a critical role in the transduction of the length signal to contractile apparatus in cardiac LDA (*Fukuda et al., 2001*; *Methawasin et al., 2014*). Increasing SL increases the passive force of cardiac muscle by stretching the titin that links the thick filament to the Z-disk and also relays its mechanical strain to the thick filament (*Linke, 2008*). A recent X-ray study on skeletal muscle showed a connection between increased thick filament stress and its transition to ON state (*Linari et al., 2015*). This mechano-sensing mechanism may also exist in cardiac muscle as the X-ray measurement of cardiac muscle has indicated a structural rearrangement within the thick filament associated with titin strain and LDA (*Ait-Mou et al., 2016*).

The results presented here, together with the previous studies described above, also demonstrated that increasing SL in cardiac muscle changes the structure of troponin in the thin filament. This length-dependent change in troponin structure is correlated with the titin strain and is present

in the absence of active force-generating myosin heads. It is not known how either titin strain or the ON state of the thick filament can be transmitted to the thin filament, although MyBP-C may be involved in this interfilament signal transmission. MyBP-C is localised to the central region of each half-thick filament (*Bennett et al., 1986*; *Luther et al., 2011*). It is bound to the thick filament via its C-terminal region (*Flashman et al., 2004*) and its N-terminal region can activate the thin filament to trigger force generation in the absence of $Ca^{2+}$ (*Herron et al., 2006*; *Kampourakis et al., 2014*; *Mun et al., 2014*). In the absence of MyBP-C, while the length-dependent increase in the maximum force generation is preserved, the length-dependent increase in $Ca^{2+}$ sensitivity is blunted (*Mamidi et al., 2014*). cMyBP-C is one of the key components in modulating cardiac contractility in a manner that depends on its phosphorylation state (*Gautel et al., 1995*; *Kunst et al., 2000*; *Pfuhl and Gautel, 2012*). Phosphorylation of MyBP-C, which inhibits the interaction of its N-terminal region with the thin filament (*Shaffer et al., 2009*; *Kampourakis et al., 2014*), has been shown to modulate LDA in cardiac muscle (*Kumar et al., 2015*; *Mamidi et al., 2016*). Thus, MyBP-C is a strong candidate for transmitting the signal of titin strain generated by SL change from the thick to the thin filament.

## Materials and methods

### Animals
All animal procedures were in accordance with Schedule 1 of the UK Animal (Scientific Procedures) Act 1986. Adult Wistar rats (200–250 g) were sacrificed by cervical dislocation. The hearts were excised and rinsed free of blood with Krebs-Henseleit solution (K3753, Sigma, St. Louis, MO) oxygenated with a carbogen mixture of 95% O2% and 5% CO2. Suitable trabeculae (free running, unbranched, diameter <250 μm) were dissected from the right ventricle in Krebs-Henseleit solution containing 25 mM 2,3-butanedione-monoxime, permeabilised in relaxing solution (see below) containing 1% Triton X-100 for 30 min, and stored in relaxing solution for experiments.

### Preparation of labelled cTnCs and cRLC
Double cysteine mutants E55C/D62C and E95C/R102C of human cTnC were obtained by site-directed mutagenesis and expressed in *E. coli*. The native cysteines C35 and C84 were replaced by serine (*Sia et al., 1997*). The difference between human and rat isoforms of cTnC is one amino acid residue at 119, Isoleucine (human) vs Methionine (rat). Ile and Met belong to the same group of amino acids with nonpolar, aliphatic side chains.

TnC was expressed, purified and labelled as described previously(*Ferguson et al., 2003*; *Sun et al., 2009*). Each pair of cysteines was cross-linked with bifunctional rhodamine (BR) to form 1:1 BR:TnC conjugates (*Corrie et al., 1998*), and purified by reverse-phase HPLC (Agilent 1200 HPLC System). Stoichiometry and specificity of BR-labelling were confirmed by electrospray mass spectrometry (Agilent 6120 Quadrupole LCMS System).

### Exchange of labelled cTnCs and cRLC into ventricular trabeculae
Endogenous cTnC was exchanged by incubation of trabeculae in relaxing solution containing 25–30 μM BR-cTnC overnight at 4°C. The fraction of cTnC replaced by BR-TnC was about 80% (*Sevrieva et al., 2014*). BSR-cRLCs were exchanged into demembranated trabeculae by extraction in CDTA-rigor solution (mM: 5 CDTA, 50 KCl, 40 Tris-HCl pH 8.4, 0.1% (v/v) Triton X-100) for 30 min followed by reconstitution with 40 mM BSR-cRLC in relaxing solution for 1 hr, replacing 30–50% of the endogenous cRLCs (*Kampourakis et al., 2014*).

Following exchange of cTnC or cRLC, the trabeculae were mounted horizontally via aluminum T-clips between a force transducer (SI-KG7, World Precision Instruments) and a fixed hook in a 60 μl trough containing relaxing solution. The experimental temperature was 20–22°C.

Experimental solutions contained 25 mM imidazole, 5 mM MgATP, 1 mM free $Mg^{2+}$, 10 mM EGTA (except pre-activating solution), 0–10 mM total calcium, 1 mM dithiothreitol and 0.1% (v/v) protease inhibitor cocktail (P8340, Sigma). Ionic strength was adjusted to 200 mM with potassium propionate; pH was 7.1 at 20°C. The concentration of free $Ca^{2+}$ was calculated using the program WinMAXC V2.5 (http://www.stanford.edu/~cpatton/maxc.html). The calculated free $[Ca^{2+}]$, expressed as pCa (i.e., $-log[Ca^{2+}]$), was in the range 1 nM (pCa 9) to 32 μM (pCa 4.5). In pre-

activating solution, [EGTA] was 0.2 mM and no calcium was added. When required, 25 µM blebbistatin (B0560, Sigma) was added from a 10 mM stock solution in DMSO.

## Measurement of probe orientation by polarised fluorescence

Measurement of fluorescence polarization was similar to previously described methods (*Sun et al., 2009*; *Sevrieva et al., 2014*). A central 0.5 mm segment of a trabecula exchanged with BR-cTnC was briefly illuminated from below with 532 nm light polarised either parallel or perpendicular to the trabecular axis. BR fluorescence at 610 nm was collected in line with the illuminating beam that was propagating perpendicular to the trabecular axis. The emitted fluorescence was then separated into parallel and perpendicular components. Together with the order parameter $<P_{2d}>$ that describes the amplitude of rapid (sub-nanosecond) probe motion and has been measured in our previous study (*Sun et al., 2009*), two independent orientation order parameters were calculated from these measured intensities: $<P_2>$ and $<P_4>$, describing the distribution of angles between BR fluorescence dipole and trabecular axis averaged over slower timescales (*Dale et al., 1999*). $<P_2>$ would be +1 if every BR dipole were parallel to the trabecular axis, and $-0.5$ if they were all perpendicular. $<P_4>$ provides higher resolution angular information, and orientational disorder decreases the absolute values of both $<P_2>$ and $<P_4>$. The detailed results are presented here only for $<P_2>$, which can be measured with greater signal-to-noise ratio. If there are multiple populations of BR dipoles with distinct orientations, the observed $<P_2>$ is linearly related to the fraction of dipoles in each population.

Each trabecular activation was preceded by a 1 min incubation in pre-activating solution. Isometric force and fluorescence intensities were measured after steady-state force had been established in each activation. Maximum force was recorded before and after each series of activations at sub-maximal [Ca$^{2+}$]. If the maximum force decreased by >15%, the trabecula was discarded. The experiment was completed when full Ca$^{2+}$-force relationships at both short and long SLs were obtained from the same trabecula. The dependence of force and $<P_2>$ on [Ca$^{2+}$] was fitted to data from individual trabeculae using nonlinear least-squares regression to the Hill equation:

$$Y = 1 / \left( 1 + 10^{n_H(pCa - pCa_{50})} \right)$$

where pCa$_{50}$ is the pCa corresponding to half-maximal change in either force or $<P_2>$, and $n_H$ is the Hill coefficient.

The steepness of the Ca$^{2+}$-force relationships in the present study ($n_H$ = ~4, *Table 1*) was less than that in a study where SL was kept constant during the contraction by adjusting overall muscle length (*Dobesh et al., 2002*). This is likely due to the end-compliance of trabecular preparations that allows SL shortening during activation (*ter Keurs et al., 1980*; *Kentish et al., 1986*). As the trabecula generates more force at higher concentrations of Ca$^{2+}$, the central sarcomeres shorten more. Consequently, the Ca$^{2+}$-force relationship shifts to the right at higher [Ca$^{2+}$] and becomes less steep. Therefore, both pCa$_{50}$ and $n_H$ of the Ca$^{2+}$-force relationship at a given SL were likely to be underestimated in fixed end conditions as in the present study. To minimise the impact of SL shortening during activation on the interpretation, only the data from experiments in which full Ca$^{2+}$-force relationships at both short and long SLs were obtained from the same trabecula were fit to the Hill equation. In addition, paired-comparisons were used throughout the present study. Thus, the differential shortening of the fixed-end contractions at the different lengths would not affect the interpretation of the results. All values are given as mean ± standard error except where noted, with $n$ representing the number of trabeculae.

## Acknowledgements

We are grateful to Dr John Corrie for the bifunctional rhodamine. This work was supported by the British Heart Foundation.

## Additional information

### Funding

| Funder | Grant reference number | Author |
| --- | --- | --- |
| British Heart Foundation | FS/15/1/31071 | Yin-Biao Sun |
| British Heart Foundation | FS/09/001/26329 | Yin-Biao Sun |

The funders had no role in study design, data collection and interpretation, or the decision to submit the work for publication.

### Author contributions

XZ, Data curation, Formal analysis, Investigation; TK, Resources, Formal analysis, Investigation; ZY, Resources; IS, Resources, Investigation; MI, Conceptualization, Supervision, Writing—review and editing; Y-BS, Conceptualization, Formal analysis, Supervision, Funding acquisition, Writing—original draft, Writing—review and editing

### Author ORCIDs

Yin-Biao Sun, http://orcid.org/0000-0002-4992-8198

### Ethics

Animal experimentation: This study was carried out in accordance with Schedule 1 of the UK Animal (Scientific Procedures) Act 1986, as approved by the King's College London Ethical Review Process committee.

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
