## [Decision Letter]

Thank you for submitting your article "Distinct contributions of the thin and thick filaments to length-dependent activation in heart muscle" for consideration by *eLife*. Your article has been favorably evaluated by Anna Akhmanova as the Senior Editor, a Reviewing Editor, and two reviewers: James Spudich (Reviewer #1) and John Solaro (Reviewer #2).

The reviewers have discussed the reviews with one another and the Reviewing Editor has drafted this decision to help you prepare a revised submission.

Summary:

The Frank-Starling law of the heart describes the relationship between cardiac stroke volume and end diastolic volume. At the cellular level, the Frank-Starling relationship implies that increasing cardiac sarcomere length results in enhanced performance of cardiac muscle cells during the subsequent contraction. Although the Frank-Starling law of the heart is one of its fundamental properties, the detailed molecular mechanism for this length-dependent activation (LDA) in heart muscle is still not understood. The work described here uses fluorescence-based methods to analyze the changes that occur in orientation of cardiac TnC on the thin filament and the regulatory light chain of the myosin heads, the molecular motor domains of myosin, as the muscle is stretched, and provides evidence that the primary reason for increased force production upon stretch is activation of myosin heads in the thick filaments.

Essential revisions:

1) The results clearly show that LDA is accompanied by distinctive structural changes in both the thin and thick filaments. The authors conclude from those structural changes that they reflect the changes in calcium sensitivity (TnC structural change) and force production (myosin head orientation changes) associated with LDA. While these are reasonable assumptions, it is not for certain that this is true. Structural changes need not reflect force changes. Thus, the authors need to point out that they are making a jump in their interpretation about this connection, and do a better job describing why they think this jump is justified and likely correct. In particular, a non-muscle biologist, not knowing the field, will not understand why a change in myosin head orientation means a likely increase in force production.

2) The paper needs to be recast to be more accessible to the broader readership of *eLife*. For example, Results: The non-muscle biologist needs to understand why the authors are concluding that the "Increase of maximum active force at longer SL is not associated with increased thin filament activation," when calcium-sensitivity is in fact increased. This section is so filled with numbers and terms that the reader finds it a bit difficult to follow the logic. Suddenly one comes to the second paragraph of the subsection “Length-dependent orientational changes of the E helix of cTnC”, which states that "For both the C and E helix probes increasing SL increased <*P_2_*> over the whole range of [Ca^2+^] in a direction characteristic of less activation, whereas up to this point calcium activation (in terms of sensitivity) is enhanced. The reader senses a disconnect. It's not clear how one knows, as mentioned in the previous paragraph, that the changes are "in the direction characteristic of a less active state (Figure 2 and Figure 3)."

Another example is in the first paragraph of the subsection “Length-dependent Ca^2+^ sensitivity of cTnC structural changes is independent of the presence of force-generating myosin heads”, where the authors state "It should be noted that blebbistatin may inhibit active force generation by stabilising the OFF structure of the myosin thick filament." We know what the OFF structure is thought to be, but no non-muscle biologist will be aware of this.

Another example is in the subsection “Increase in maximal force in LDA is due to a change in thick filament structure”, which says "increasing SL from 1.9 to 2.3 μm increased <*P_2_*> for the BSR-cRLC-BC probe in the direction associated with activation." How does one know that this direction is associated with activation? Just being more perpendicular to the thin and thick filaments doesn't necessarily mean more activation.

3) The authors need to briefly discuss limitations of their approach. Of significance is the exclusive use of steady state measurements, whereas the system operates dynamically during the beats of the heart.

---

## [Author Response]

*Essential revisions:*

*1) The results clearly show that LDA is accompanied by distinctive structural changes in both the thin and thick filaments. The authors conclude from those structural changes that they reflect the changes in calcium sensitivity (TnC structural change) and force production (myosin head orientation changes) associated with LDA. While these are reasonable assumptions, it is not for certain that this is true. Structural changes need not reflect force changes. Thus, the authors need to point out that they are making a jump in their interpretation about this connection, and do a better job describing why they think this jump is justified and likely correct. In particular, a non-muscle biologist, not knowing the field, will not understand why a change in myosin head orientation means a likely increase in force production.*

The relationship between the conformation of myosin motor, the regulatory state of the thick filament, and active force generation has now been explained more fully (subsection “Dual filament regulation of length-dependent activation in cardiac muscle”, first paragraph). The TnC structural changes in the activation of heart muscle cells have also been stated clearly for the broader audience of *eLife* (subsection “Increase of maximum active force at longer SL is not associated with increased thin filament activation”, second paragraph).

*2) The paper needs to be recast to be more accessible to the broader readership of eLife. For example, Results: The non-muscle biologist needs to understand why the authors are concluding that the "Increase of maximum active force at longer SL is not associated with increased thin filament activation," when calcium-sensitivity is in fact increased.*

This has now been explained more fully in a separate paragraph (subsection “Increase in maximal force in LDA is linked to a change in thick filament structure”).

*This section is so filled with numbers and terms that the reader finds it a bit difficult to follow the logic. Suddenly one comes to the second paragraph of the subsection “Length-dependent orientational changes of the E helix of cTnC”, which states that "For both the C and E helix probes increasing SL increased <P_2_> over the whole range of [Ca^2+^] in a direction characteristic of less activation, whereas up to this point calcium activation (in terms of sensitivity) is enhanced. The reader senses a disconnect. It's not clear how one knows, as mentioned in the previous paragraph, that the changes are "in the direction characteristic of a less active state (Figure 2 and Figure 3)."*

The Results section has now been revamped to make it more accessible to the broader readership of *eLife*.

*Another example is in the first paragraph of the subsection “Length-dependent Ca^2+^ sensitivity of cTnC structural changes is independent of the presence of force-generating myosin heads”, where the authors state "It should be noted that blebbistatin may inhibit active force generation by stabilising the OFF structure of the myosin thick filament." We know what the OFF structure is thought to be, but no non-muscle biologist will be aware of this.*

The OFF structure of the myosin thick filament is now been briefly explained (subsection “Length-dependent Ca^2+^ sensitivity of cTnC structural changes is independent of the presence of force-generating myosin heads”, first paragraph).

*Another example is in the subsection “Increase in maximal force in LDA is due to a change in thick filament structure”, which says "increasing SL from 1.9 to 2.3 μm increased <P_2_> for the BSR-cRLC-BC probe in the direction associated with activation." How does one know that this direction is associated with activation? Just being more perpendicular to the thin and thick filaments doesn't necessarily mean more activation.*

This statement has been re-formulated to make it clearer that the effect of increasing SL on the <*P*_2_> for the BSR-cRLC-BC probe (subsection “Increase in maximal force in LDA is linked to a change in thick filament structure”).

*3) The authors need to briefly discuss limitations of their approach. Of significance is the exclusive use of steady state measurements, whereas the system operates dynamically during the beats of the heart.*

The limitations of the present results from steady state measurements are now stated at the start of Discussion. Also, a paragraph related to the limitations of the present technique has been moved from the Results to the Discussion (second paragraph).